# Autoreactive CD8 T cells in NOD mice exhibit phenotypic heterogeneity but restricted TCR gene usage

Moujtaba Y Kasmani[1,2,*] ⓘ, Ashley E Ciecko[3,4,*], Ashley K Brown[1,2], Galina Petrova[3], Jack Gorski[1,2], Yi-Guang Chen[1,3,4] ⓘ, Weiguo Cui[1,2] ⓘ

**Type 1 diabetes (T1D) is an autoimmune disorder defined by CD8 T cell–mediated destruction of pancreatic $\beta$ cells. We have previously shown that diabetogenic CD8 T cells in the islets of non-obese diabetic mice are phenotypically heterogeneous, but clonal heterogeneity remains relatively unexplored. Here, we use paired single-cell RNA and T-cell receptor sequencing (scRNA-seq and scTCR-seq) to characterize autoreactive CD8 T cells from the islets and spleens of non-obese diabetic mice. scTCR-seq demonstrates that CD8 T cells targeting the immunodominant $\beta$-cell epitope IGRP$_{206-214}$ exhibit restricted TCR gene usage. scRNA-seq identifies six clusters of autoreactive CD8 T cells in the islets and six in the spleen, including memory and exhausted cells. Clonal overlap between IGRP$_{206-214}$–reactive CD8 T cells in the islets and spleen suggests these cells may circulate between the islets and periphery. Finally, we identify correlations between TCR genes and T-cell clonal expansion and effector fate. Collectively, our work demonstrates that IGRP$_{206-214}$–specific CD8 T cells are phenotypically heterogeneous but clonally restricted, raising the possibility of selectively targeting these TCR structures for therapeutic benefit.**

## Introduction

Type 1 diabetes (T1D) is an autoimmune condition that results in destruction of insulin-producing $\beta$ cells in the pancreatic islets, thereby leaving affected patients dependent on exogenous insulin therapy (Bluestone et al, 2010). T1D is thought to be primarily mediated by T cells, with both CD4 and CD8 T cells involved in human disease and the non-obese diabetic (NOD) mouse model (Makino et al, 1980; Mullen, 2017). Over the last three decades, CD8 T cells have been studied more extensively in the context of chronic or relapsing inflammatory conditions such as chronic viral infections (Moskophidis et al, 1993; Zajac et al, 1998; Wherry et al, 2003),

cancers (Ahmadzadeh et al, 2009; Siddiqui et al, 2019), and autoimmune conditions including T1D (McKinney et al, 2015; Long et al, 2016; Hu et al, 2020). Application of high-throughput sequencing, including single-cell RNA sequencing (scRNA-seq), to models of these various inflammatory states by our laboratory and others has shown previously unappreciated heterogeneity in the CD8 T-cell compartment, demonstrating the coexistence of progenitor-, effector-, and exhausted-like CD8 T cells in these conditions (He et al, 2016; Im et al, 2016; Chen et al, 2019, 2021; Hudson et al, 2019; Zander et al, 2019; Abdelsamed et al, 2020; Beltra et al, 2020; Sandu et al, 2020; Wiedeman et al, 2020; Zakharov et al, 2020; Ciecko et al, 2021; Connolly et al, 2021; Gearty et al, 2022).

Some works have suggested that an increased exhausted state, rather than a cytolytic effector state, in the CD8 T-cell compartment may be protective in autoimmune diseases (McKinney et al, 2015). In line with this, a recent study of T1D patients demonstrated a correlation between increased frequencies of TIGIT$^+$ EOMES$^+$ islet–specific CD8 T cells and slower disease progression (Wiedeman et al, 2020). These cells were found to exhibit an exhausted-like phenotype, similar to the increased expression of EOMES observed in exhausted CD8 T cells in murine models of chronic viral infection (Paley et al, 2012; Zander et al, 2019). Despite the important clinical impact and novel findings of this work, the data are limited as CD8 T cells were collected from peripheral blood, raising the question of whether CD8 T cells detected peripherally provide an accurate sampling of the autoreactive lymphocyte pool in the pancreatic islets.

Further investigation into the clonal landscape of CD8 T cells, in which a T-cell clone is defined as a group of T cells arising from the same naïve T cell and sharing the same unique T-cell receptor generated by V(D)J recombination (Oettinger et al, 1990), may provide further insight into the pathogenesis of insulitis. Mathematical estimates of TCR diversity have suggested that $10^{15}$–$10^{20}$ unique TCRs are capable of being generated by random rearrangement (Davis & Bjorkman, 1988; Lieber, 1991), although individual organisms have far fewer naïve T cells in reality. Recent work from our laboratory using single-cell TCR sequencing (scTCR-seq)

[1]Department of Microbiology and Immunology, Medical College of Wisconsin, Milwaukee, WI, USA    [2]Blood Research Institute, Versiti Wisconsin, Milwaukee, WI, USA    [3]Department of Pediatrics, Medical College of Wisconsin, Milwaukee, WI, USA    [4]Max McGee National Research Center for Juvenile Diabetes, Medical College of Wisconsin, Milwaukee, WI, USA

Correspondence: wecui@mcw.edu; yichen@mcw.edu
*Moujtaba Y Kasmani and Ashley E Ciecko are co-first authors.

has demonstrated that virus-specific CD4 T cells during acute infection are highly clonally heterogeneous, with variable V(D)J gene usage detected and the vast majority of clones not shared among any of five mice studied (Khatun et al, 2021). However, CD8 T-cell clonal heterogeneity in the context of T1D remains understudied at the single-cell level. Given recent advancements in the use of high-throughput sequencing to better define T-cell phenotypes and clonotypes, and to build upon recent work from our laboratory and others demonstrating that islet-specific CD8 T cells are composed of self-renewing progenitor cells that give rise to effector cells (Ciecko et al, 2021; Gearty et al, 2022), we decided to investigate the phenotypic and clonal landscapes of CD8 T cells from the islets and spleens of prediabetic NOD mice using paired scRNA-seq and scTCR-seq with a particular focus on CD8 T cells targeting the immunodominant autoantigen $IGRP_{206-214}$.

# Results

### Single-cell T-cell receptor sequencing reveals vast clonal heterogeneity among $IGRP_{206-214}$–reactive CD8 T cells despite restricted TCR gene usage

To investigate the phenotypic and clonal landscapes of CD8 T cells in detail, we subjected CD8 T cells harvested from NOD mice to paired single-cell RNA sequencing (scRNA-seq) and single-cell T-cell receptor sequencing (scTCR-seq). We performed two independent experiments in which we harvested and pooled spleens and islets from 10 individual 10- to 15-wk-old prediabetic NOD females and sorted $IGRP_{206-214}$ tetramer–positive CD8 T cells from both tissues as well as tetramer-negative CD8 T cells from islets (Figs 1A and S1A). Performing TCR sequencing at the single-cell level allowed us to define T-cell clones based on paired TCR α and β chain complementarity determining region 3 (CDR3) sequences at the nucleotide level. This approach provides a strong definition of T-cell clonality as the CDR3 region, the most variable portion of the TCR and the region that directly contacts peptides presented on MHC (Cole et al, 2014), includes germline information from V(D)J genes as well as N nucleotides randomly inserted into the junctions between these genes. Of 25,209 total CD8 T cells sequenced, we recovered paired TCR α and β chain sequences from 15,226 cells (60.4%).

We used a Venn diagram to compare overlap of clones with at least two constituent CD8 T cells recovered from the islets in each of our two experiments (Fig 1B). We defined a clone as a group of T cells sharing the same paired TCR α and β chain CDR3 nucleotide sequence. Five of 81 $IGRP_{206-214}$ tetramer–positive islet clones were shared between the two groups of mice (6.2% overlap), as were two out of 724 $IGRP_{206-214}$ tetramer–negative clones (0.3% overlap). A total of five clones were also shared between $IGRP_{206-214}$ tetramer–positive and –negative cells within individual experiments, likely the result of inherent limitations to the accuracy of FACS sorting. When examining clones isolated from the spleen, we found four of 111 clones shared between experiments (3.6% overlap) (Fig S2A). However, when clones from islets and spleens were combined, we found that only seven of 840 clones were

shared between the two groups of 10 pooled mice (0.8% overlap) (Fig S2B). Moreover, if the same group of cells was analyzed with clones being defined using only TCR α or TCR β chain CDR3 nucleotide sequences, rather than paired CDR3 sequences, higher frequencies of overlapping clones (23/740 = 3.1% overlap and 11/784 = 1.4% overlap, respectively) were found (Fig S2B), thus emphasizing the increased resolution afforded by scTCR-seq compared to bulk TCR sequencing. These data suggest that clonal overlap of autoreactive CD8 T cells between any two genetically identical NOD mice is extremely low; this validates mathematical predictions of V(D)J recombination diversity (Davis & Bjorkman, 1988; Lieber, 1991), which suggest that $10^{15}$ to $10^{20}$ TCRs may be randomly generated by an organism, and our laboratory's prior work on CD4 T-cell clonal diversity in acute viral infection (Khatun et al, 2021). Despite a relatively high level of diversity, there was higher clonal overlap between $IGRP_{206-214}$–reactive clones compared with other CD8 T-cell clones, as visualized by an UpSet diagram of clonal overlap (Fig S2C). Of note, there is a large number of the same $IGRP_{206-214}$–reactive clones present in both islets and spleens in each of the two experiments, suggesting continuous trafficking between islets and peripheral lymphoid organs.

$IGRP_{206-214}$ tetramer–positive CD8 T cells, both in the islets and in the spleen, also exhibited high levels of clonal expansion compared with tetramer-negative cells (Fig 1C). When examining clones with at least two constituent cells, an average tetramer-positive clone consisted of ~24 cells, whereas an average tetramer-negative clone consisted of approximately six cells (Fig S2D). Despite this, we still detected several large $IGRP_{206-214}$ tetramer–negative clones, including one clone with more than 200 cells and nine clones with more than 50 cells. These data reinforce the fact that although IGRP is the major autoantigen in the islets in NOD mice during the progression of T1D (Lieberman et al, 2003; Trudeau et al, 2003), multiple epitopes such as insulin and glutamate decarboxylase are targeted by CD8 T cells in both NOD mice and human T1D patients (Amdare et al, 2021).

Intrigued by our findings that autoreactive CD8 T-cell clones exhibit low level of clonal overlap between organisms, we used chord diagrams to visualize V-J gene pairings of CD8 T cells recovered from the islets. D genes were excluded from this analysis as these are only found in the TCR β chain and the mouse genome only contains two D genes (TRBD1 and *TRBD2*). Although we expected to see highly variable V-J gene pairings similar to what we previously observed in CD4 T cells in the setting of acute viral infection (Khatun et al, 2021), we found different trends between $IGRP_{206-214}$–reactive and –nonreactive CD8 T cells (Fig 1C). $IGRP_{206-214}$–nonreactive (tetramer-negative) CD8 T cells from the islets exhibited highly diverse V-J gene pairings in the TCR α and β chain in both experiments, as expected (Fig 1D). However, $IGRP_{206-214}$–reactive (tetramer-positive) cells had highly restricted V-J pairings in both the TCR α and β chains. Specifically, over 99% of these cells used TRAV16N and 86–92% of total cells paired this V gene with TRAJ42 to form a TCR α chain. Approximately 90% of tetramer-positive cells expressed TRBV13-3 and more than 90% used either TRBJ2-4, TRBJ2-7, or TRBJ2-2 for the β chain J gene. Similar trends were found in tetramer-positive CD8 T cells recovered from the spleen, with more than 97% of cells expressing TRAV16N; 79–94% TRAJ42; 81–85% TRBV13-3; and 90–92% TRBJ2-4, TRBJ2-7, or TRBJ2-2.

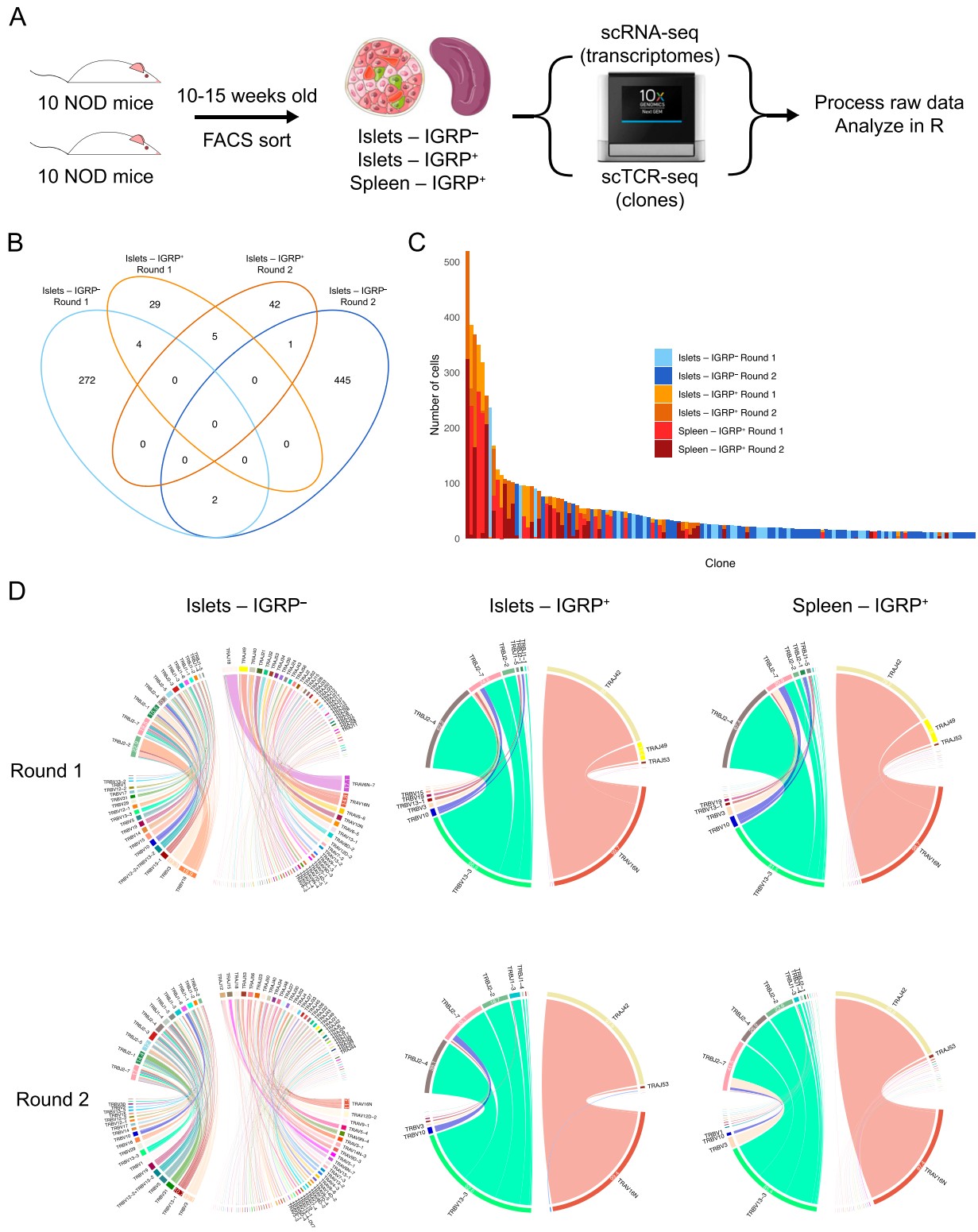

**Figure 1. Single-cell T-cell receptor sequencing reveals vast clonal heterogeneity among IGRP$_{206-214}$–reactive CD8 T cells despite restricted TCR gene usage.**
**(A)** Experimental design of paired scRNA-seq and scTCR-seq experiments performed on CD8 T cells isolated from the islets and spleens of non-obese diabetic mice.
**(B)** Venn diagram showing clonal overlap among CD8 T cell clones isolated from islets. Clones are defined as groups of CD8 T cells sharing the same paired TCR $\alpha$ and $\beta$ chain CDR3 nucleotide sequences. Singlet clones were excluded from this analysis as, by definition, they do not overlap between samples. **(C)** Bar plot showing size and sample origin of clones with at least 10 constituent cells. Each bar represents one clone, color denotes sample identity. **(D)** Chord diagrams showing V and J gene pairings within TCR $\alpha$ and $\beta$ chains of IGRP$_{206-214}$–reactive cells from islets and spleens and IGRP$_{206-214}$–nonreactive cells from islets. Colors denote individual TCR genes,

## Single-cell RNA sequencing elucidates the phenotypic heterogeneity of diabetogenic CD8 T cells in the islets and spleen

To investigate the phenotypes of CD8 T cells in T1D we performed scRNA-seq of the IGRP$_{206-214}$ tetramer–positive and tetramer-negative CD8 T cells isolated from the islets and spleens of NOD mice. To prevent overfitting phenotypes, sctransform integration and dimensionality reduction using Seurat's UMAP algorithm was performed separately on samples from islets and spleens (see the Materials and Methods section). Analysis of integrated tetramer-positive and tetramer-negative cells from the islets revealed six clusters (Fig 2A). Identities of these CD8 T-cell clusters were determined using differentially expressed genes (Fig 2B). Bystander cells expressed *Sell* (encodes L-selectin) and had the lowest expression of *Cd44*; progenitor cells expressed *Tcf7* and *Slamf6* (encodes Ly108). Effector cells expressed the cytotoxic molecule *Gzmb*, the chemokine receptor *Cxcr6*, and coinhibitory receptors associated with T-cell exhaustion such as *Pdcd1* (encodes PD-1) and *Lag3*. Recently activated CD8 T cells were identified by high expression of *Cd69*, an early marker of T-cell activation (Yokoyama et al, 1988), and *Nr4a1* (encodes Nur77, a transcription factor [TF] downstream of TCR signaling) (Moran et al, 2011). Mitotic cells highly expressed *Top2a* (encodes DNA topoisomerase II alpha). Finally, anergic cells up-regulated *Cblb* (encodes a ubiquitin kinase that inhibits TCR signaling) (Bachmaier et al, 2000; Chiang et al, 2000; Jeon et al, 2004); *Itpkb* (encodes inositol triphosphate 3 kinase beta), which restrains T-cell cytokine production (Pouillon et al, 2013) and inhibits thymic TCR signaling (Westernberg et al, 2016); and *Dgka* (encodes diacylglycerol kinase alpha), a kinase that inhibits TCR signaling to promote anergy (Olenchock et al, 2006).

We also examined the relative frequencies of these clusters in islet tetramer-positive and -negative samples and found that bystander cells were enriched in tetramer-negative samples in both independent experiments (Figs S3A and 2C). Conversely, effector, mitotic, and anergic cells were enriched in tetramer-positive samples in both experiments. These data suggest that IGRP$_{206-214}$–reactive effector CD8 T cells dominate the islet autoimmune response at 10–15 wk of age in NOD mice as they are more proliferative and able to adopt a cytotoxic phenotype. These cells may also be more prone to developing an anergic phenotype as a result of overstimulation. Development of anergy would also be expected to limit CD8 T cell clonal proliferative capacity. To test this, we used our scTCR-seq data to break down each islet cluster by frequency of clone size (Fig S3A) Indeed, IGRP$_{206-214}$–nonreactive clones in the anergic cluster, as well as those in the progenitor and recently activated clusters, had relatively low expansion, with ~80% being singlet clones. IGRP$_{206-214}$–nonreactive clones with bystander constituent cells were even more extreme in their lack of expansion, with almost 99% being singlet clones. In contrast, IGRP$_{206-214}$–nonreactive effector and mitotic clones proliferated much more, with ~60% of clones being singlets and 25% having at least three cells. IGRP$_{206-214}$–reactive clones, on the other hand, were all more expanded than their IGRP$_{206-214}$–nonreactive counterparts.

Despite this, similar trends among clusters could be seen, with effector and mitotic cells being more expanded than other clusters. IGRP$_{206-214}$–reactive bystander cells exhibited the greatest difference in clonal expansion compared with IGRP$_{206-214}$–nonreactive bystander cells, suggesting that these cells may have already undergone some degree of activation due to the immunodominance of IGRP$_{206-214}$ as an autoantigen in the islets.

We have previously used scRNA-seq to identify four major subsets of CD8 T cells (bystander, progenitor, effector, and mitotic) in the islets of NOD mice at 7 and 14 wk of age (Ciecko et al, 2021); our previous work, as well as that from other groups (Hu et al, 2020; Gearty et al, 2022), has shown that progenitor cells give rise to effector cells that then mediate insulitis. Our current study has much higher resolution as we were able to collect and pool a much greater number of cells for scRNA-seq, allowing us to identify greater heterogeneity in the diabetogenic CD8 T-cell pool compared with our previous work (Ciecko et al, 2021). To compare the cells in our current data with our previously identified subsets, we generated module scores from differentially expressed genes of the four subsets in our previous publication (Fig 2D). This analysis showed that the current bystander, progenitor, effector, and mitotic clusters scored highly for their respective module scores derived from our previous data. In addition, our newly identified recently activated and anergic clusters both scored highly for the progenitor module score, suggesting that these two clusters may have been part of our older progenitor cluster but were not identified because of limited scRNA-seq resolution at the time. Given that lymphocytic choriomeningitis virus Clone 13 (LCMV Cl13) chronic infection system is a common model system of persistent antigen stimulation and CD8 T-cell exhaustion, we also used module scores derived from one of our laboratory's prior publications (Zander et al, 2019) to score our islet CD8 T-cell clusters (Fig 2E). This analysis confirmed that progenitor, bystander, and recently activated cells are more quiescent because they had the highest Cl13 Progenitor module scores. Autoreactive effector cells scored highly for Cl13 Effector modules as expected but also had high Cl13 Exhausted module scores in line with their expression of inhibitory coreceptors (Fig 2B). Mitotic cells had a reasonably high Cl13 Effector module score and an intermediate Exhausted module score. Anergic cells scored relatively low in all categories but had an especially low Effector module score. Collectively, these analyses suggest that CD8 T cells in the islets of NOD mice are phenotypically heterogeneous. Although we did not detect a discrete exhausted CD8 T-cell population in the islets, we did find that autoreactive effector cells express coinhibitory receptors and have a similar gene expression profile to both Effector and Exhausted cells from LCMV Cl13 chronic infection. Overall, our results concerning islet CD8 T-cell differentiation states is consistent with a recent report (Zakharov et al, 2020) but in addition reveals an anergic population, as supported by its gene expression profile and reduced clonal expansion.

We then turned our attention to IGRP$_{206-214}$–reactive CD8 T cells isolated from the spleens of NOD mice. UMAP projection with unbiased clustering revealed 6 clusters (Fig 2F): effector memory,

chord width correlates with frequency of cells using a gene. Gene usage frequencies are shown on the edge of each chord, with frequencies less than 10% not shown for visual clarity. TCR gene names are shown outside each chord, with genes used by less than 2% of cells not shown for visual clarity.

**Figure 2.   Single-cell RNA sequencing elucidates the phenotypic heterogeneity of diabetogenic CD8 T cells in the islets and spleen.**
**(A)** UMAP plot showing cluster identities of individual CD8 T cells isolated from islets. Each dot represents one cell, colors denote cluster identity as determined by unsupervised clustering. **(B)** Dot plot showing expression of key marker genes among different CD8 T-cell clusters from islets. Dot size indicates the frequency of cells in a cluster expressing a particular gene, dot colors denotes average expression of that gene. **(C)** Bar graph showing islet cluster distribution frequencies of each sample.
**(A)** Colors correspond to the clusters shown in (A). **(A, D)** Dot plot showing module scores within each cluster from (A) of the top 100 differentially expressed genes in the four CD8 T cell clusters previously identified in non-obese diabetic islets (Ciecko et al, 2021). **(D, E)** As in (D), but using the top 100 differentially expressed genes among

*AY036118*[+], central memory, *Cx3cr1*[+] effector, exhausted, and anergic CD8 T cells. As before, clusters were defined based on expression of key marker genes (Fig 2G). Effector memory and *AY036118*[+] cells had a very similar gene expression profile and up-regulated memory genes such as *Il7r* (encodes IL-7R), *Tcf7* (encodes TCF-1), and *Slamf6* (encodes Ly108) as well as effector genes such as *Tbx21* (encodes T-bet) and *Klrg1*; however, only the latter cluster up-regulated *AY036118*, which encodes the transcriptional repressor ERF1 (Verykokakis et al, 2007) and was recently reported to be up-regulated in CD8 T cells with repressed activation in a lung cancer model (Burger et al, 2021). Central memory cells highly expressed the aforementioned memory markers as well as *Sell* (encodes L-selectin) and *Ccr7*. Splenic *Cx3cr1*[+] effector cells expressed *Gzmb* and *Klrg1*, similar to effector cells from the islets, but critically had minimal expression of coinhibitory receptor genes such as *Pdcd1* and *Lag3*. Instead, these molecules were up-regulated by Exhausted cells, which also expressed the chemokine receptor *Cxcr6* and the exhaustion-associated TF *Eomes* (Paley et al, 2012; Long et al, 2016; Siddiqui et al, 2019). Of note, the effector memory and *AY036118*[+] clusters also expressed *Eomes*, albeit at a lower level, likely due to the fact that Eomes is involved in CD8 T cell memory formation (Banerjee et al, 2010; Kaech & Cui, 2012). Finally, splenic anergic cells were similar to their counterparts in the islets as they also up-regulated *Cblb*, *Itpkb*, and *Dgka*, genes encoding TCR inhibitory molecules. Scoring of spleen clusters using LCMV Cl13 module scores demonstrated expected trends (Fig 2H). The effector memory and *AY036118*[+] clusters exhibited intermediate scores of all three Cl13 modules with a slightly higher Progenitor module score, as we have shown that Cl13 Progenitor cells are similar to memory precursor cells from acute viral infection (Chen et al, 2021). Central memory cells had a high Progenitor module score, *Cx3cr1+* effector cells had a high Effector module score, and exhausted cells had a high Cl13 Exhausted module score. Anergic cells in the spleen had relatively low scores but scored slightly best for the Effector module, unlike islet anergic cells. Clonal expansion in splenic CD8 T cell clusters was more restrained compared with IGRP$_{206-214}$–reactive islet clusters (Fig S3B): Effector memory, *AY036118*[+], *Cx3cr1*[+] effector, and exhausted clones had up to 50% of non-singlet clones. This was in line with IGRP$_{206-214}$–reactive progenitor cells, which also have a memory-like phenotype. Only two clones were covered from the very small splenic anergic cluster, one singlet and one doublet clone. The major outlier was the central memory cluster, which was almost entirely (>97%) composed of singlet clones. Although it may be possible that some of the cells in this cluster were actually naïve cells, given that naïve and memory T cells share many transcriptomic and epigenetic similarities (Schauder et al, 2021), the splenic CD8 T cells we sequenced were sorted to be CD44[+] (Fig S1B); these data therefore suggest that central memory CD8 T cells may undergo clonal contraction. In general, splenic CD8 T cell clusters were less clonally expanded than those in the islets, as would be expected given that the islets are the major source of autoantigens in T1D.

Finally, we leveraged the DoRothEA curated gene regulatory network (regulon) resource (Garcia-Alonso et al, 2019; Holland et al, 2020a, 2020b) to predict activity of the top 50 most globally variable TFs among clusters of CD8 T cells in our scRNA-seq data (Fig S3C). Regulon analysis circumvents difficulties detecting TFs by scRNA-seq because of low RNA copy numbers by also analyzing coexpression of TF target genes. These regulons, consisting of TFs and their targets, are then scored for putative activity. Islet mitotic cells were excluded from this analysis as their extremely high expression of cell cycle-related TFs skews scaled heat maps. Regulon analysis predicted enriched activity of TCR signaling-induced TFs such as Nfkb1 and Jun (Courtney et al, 2018) in recently activated islet CD8 T cells as would be expected. In addition, recently activated, bystander, and progenitor cells in the islets all had high predicted activity of Stat3, a TF downstream of IL-27 signaling; this is supported by our recent work showing that IL-27 signaling promotes progenitor to effector CD8 T-cell differentiation in the islets of NOD mice (Ciecko et al, 2021). Conversely, islet effector cells exhibited increased activity of TFs associated with aerobic glycolysis, such as Hif1a (Shi et al, 2011) and Foxk2 (Sukonina et al, 2019); aerobic glycolysis is crucial for the effector function of activated T cells (Gupta et al, 2020), and we have found that *Hif1a* is also up-regulated in effector-like tumor-infiltrating lymphocytes (Topchyan et al, 2021).

Interestingly, DoRothEA analysis predicted that the TF Zeb2, known to promote the function of long-lived effector cells (LLECs) and effector memory CD8 T cells in acute viral infection (Renkema et al, 2020), is active in splenic effector memory, *AY036118*[+], and *Cx3cr1*[+] effector cells. Moreover, although Zeb2 is also critical for short-lived effector cell (SLEC) function in acute viral infection (Dominguez et al, 2015; Omilusik et al, 2015; Guan et al, 2018), Zeb2 activity was predicted to occur at intermediate levels in recently activated, progenitor, and effector CD8 T cells in the islets. This analysis suggests that Zeb2 may differentially control the effector and memory potential of diabetogenic CD8 T cells in the islets and spleen. Regulon analysis also predicted that interferon response factor 1 (IRF1), a TF activated downstream of type I IFN signaling (Li et al, 1996), is the most active regulon in exhausted CD8 T cells in the spleen. In models of chronic viral infection, blocking type I IFN signaling promotes viral control, likely due to reduced CD8 T cell exhaustion (Teijaro et al, 2013; Wilson et al, 2013; Wu et al, 2016); similar mechanisms may therefore regulate the development of exhausted CD8 T cells in T1D. Overall, this analysis showed similarity in TF activity among the different populations within the islets compared with the spleen, except for anergic cells which display similar TF activity regardless of tissue localization. We have previously demonstrated the power of regulon analysis as a tool to investigate the gene regulatory networks that govern CD8 T cell differentiation and function in chronic viral infection (Chen et al, 2021); application of such analytical pipelines to T1D may provide further insight into how diabetogenic CD8 T cells are regulated at the transcriptomic and epigenetic levels.

---

previously identified splenic CD8 T cell clusters from LCMV Clone 13 infection (Zander et al, 2019). **(A, F)** As in (A), but showing IGRP$_{206-214}$–reactive CD8 T cells isolated from spleens of non-obese diabetic mice. **(B, G)** As in (B), but showing gene expression among CD8 T cell clusters from spleen. **(E, H)** As in (E), but showing CD8 T cell clusters from spleens.

## IGRP$_{206-214}$–reactive CD8 T cells exhibit characteristics of increased antigen exposure compared with IGRP$_{206-214}$–nonreactive CD8 T cells

Given that IGRP$_{206-214}$ is the predominant islet autoantigen in NOD mice (Lieberman et al, 2003; Trudeau et al, 2003), we decided to examine phenotypic differences between IGRP$_{206-214}$ tetramer–positive (red) and -negative (blue) CD8 T cells in the islets. To do so, we examined differential expression of genes related to antigen stimulation within each islet cluster (Fig 3A). We found that the coinhibitory receptors *Pdcd1*, *Tigit*, and *Lag3* were all up-regulated in IGRP$_{206-214}$–reactive CD8 T cells, as were the exhaustion-related TFs *Tox* (Alfei et al, 2019; Khan et al, 2019; Scott et al, 2019; Seo et al, 2019; Yao et al, 2019) and *Nr4a2* (Zander et al, 2019). Interestingly, up-regulation of these genes was not exclusive to less quiescent clusters such as effector cells and mitotic cells; IGRP$_{206-214}$–reactive progenitor and bystander cells, as well as recently activated and anergic cells, up-regulated these genes compared with their IGRP$_{206-214}$–nonreactive counterparts; this supports our data showing that IGRP$_{206-214}$–reactive bystander cells are clonally expanded (Fig S3B) and further suggests that they may be antigen-experienced to some degree. Given their expression of genes encoding inhibitory coreceptors and *Tox*, IGRP$_{206-214}$–reactive bystander cells are likely distinct from IGRP$_{206-214}$–nonreactive bystander cells, which do not demonstrate expression of these activation markers and are clonally non-expanded (Fig S3B). Moreover, *Tox* expression was also observed in IGRP$_{206-214}$–reactive CD8 T cells from the spleen, including in the memory-like Effector memory and *AY036118*+ clusters, albeit at lower levels than in exhausted or anergic splenocytes (Fig 3B). This observation is in line with several recent studies that have shown that a "molecular scar" of TOX expression and epigenetic accessibility is present in memory CD8 T cells generated in settings of persistent antigen exposure, such as chronic viral infection or T1D (Wieland et al, 2017; Abdelsamed et al, 2020; Charmoy et al, 2021; Heim et al, 2021; Hensel et al, 2021; Tonnerre et al, 2021; Yates et al, 2021). Curiously, the central memory and *Cx3cr1*+ effector cells in the spleen had minimal *Tox* expression despite being activated and therefore presumably antigen-experienced. We then quantified exhaustion more broadly by plotting module scores of our LCMV Cl13 Exhausted gene list (Fig 2E) among IGRP$_{206-214}$–reactive and –nonreactive islet CD8 T cells (Fig 3C). This analysis demonstrated a statistically significant increase in Cl13 Exhausted module scores in IGRP$_{206-214}$–reactive compared with –nonreactive CD8 T cells in all clusters except the mitotic cluster. Conversely, IGRP$_{206-214}$–nonreactive CD8 T cells tended to up-regulate memory- and stem-like genes such as *Slamf6*, *Ccr7*, *Il7r*, *Tcf7*, and *Klf2* compared with their IGRP$_{206-214}$–reactive counterparts (Fig 3D). Expression of the first four genes was low in the effector and mitotic clusters, but even these clusters demonstrated differential expression of *Klf2*. Accordingly, a module of differentially expressed genes from LCMV Cl13 Progenitor cells was significantly enriched in IGRP$_{206-214}$–nonreactive cells in every cluster (Fig 3E). These analyses suggest that autoreactive CD8 T cells specific for epitopes other than IGRP$_{206-214}$ adopt a more quiescent transcriptional profile during the later stages of insulitis (10–15 wk of age) in NOD mice despite the fact that they do not form distinct phenotypes compared with IGRP$_{206-214}$–reactive diabetogenic CD8 T cells.

## IGRP$_{206-214}$–reactive CD8 T-cell phenotype correlates with T-cell receptor gene usage

Given that IGRP$_{206-214}$–reactive CD8 T cells exhibit striking clonal diversity (Fig 1B) despite restricted TCR gene usage (Fig 1D) and that diabetogenic CD8 T cells from transgenic mice or culture systems are known to have limited TCR diversity (Verdaguer et al, 1996; DiLorenzo et al, 1998), we hypothesized that most TCR sequences from IGRP$_{206-214}$–reactive CD8 T cells would bear similar sequences. To test this hypothesis, we performed CDR3 amino acid motif analysis using TCRdist (Dash et al, 2017). Paired TCR *α* and *β* chain motif analysis of all IGRP$_{206-214}$–reactive clones with at least two constituent cells revealed strong motif consensus between large numbers of IGRP$_{206-214}$–reactive CD8 T-cell clones (Fig 4A). Of the 192 clones analyzed, 135 clones (70.3%) exhibited an "RDSG" amino acid motif in the *α* chain CDR3 (Fig S4A). 42 clones (21.9%) also demonstrated an "SSDP" motif at the beginning of the *β* chain CDR3 (Fig S4B). Other major *β* chain motifs included an "S-DW" motif found in 17 clones (8.9%) and a "GDN" motif found in 16 clones (8.3%) (Fig S4B). Conversely, IGRP$_{206-214}$–nonreactive clones had a broad range of CDR3 motifs, with consensus motifs occurring among a far lower frequency of clones compared with IGRP$_{206-214}$–reactive cones (Fig 4B). Interestingly, 10 of 724 clones analyzed (1.4%) had a fairly conserved "E-RGS" motif in the *α* chain CDR3 paired with an "RGQSN" motif in the *β* chain CDR3. These 10 clones also exhibited restricted TCR gene usage (TRAV4D-3, TRAJ18, TRVB13-1, and TRJB1-4/1-1); by analogy to similar characteristics found in IGRP$_{206-214}$–reactive clones, these 10 IGRP$_{206-214}$–nonreactive clones may all be specific for the same autoantigen. Returning to our IGRP$_{206-214}$–reactive motif analysis (Fig 4A), we noticed that the three TCR *β* chain motifs primarily used the 2-2, 2–4, or 2–7 variant of the TCR *β* chain J gene, in line with these three genes collectively being used by ~90% of IGRP$_{206-214}$–reactive CD8 T cells (Fig 1D). We began to investigate differences imparted by differential TCR *β* J gene usage by quantifying TCR signaling within islet CD8 T cells. We used the "KEGG TCR Signaling Pathway" geneset (GSEA systematic name M9904) to score islet IGRP$_{206-214}$–reactive CD8 T cells using Seurat's Module Score function. Doing so showed that IGRP$_{206-214}$–reactive cells using TRBJ2-2 express significantly higher levels of genes involved in TCR signaling than those cells using TRBJ2-4 or TRBJ2-7 (Fig S4C), suggesting that the TCR structure encoded by TRBJ2-2 may promote TCR signaling upon binding to H-2K$^d$:IGRP$_{206-214}$. Given that TCR motifs correlate with cell fate (Khatun et al, 2021) and that TCR signaling has an impact on T-cell expansion (Ozga et al, 2016), we wondered if use of these TCR *β* chain J genes might correlate with clonal expansion. Using logistic regression analysis, we found that IGRP$_{206-214}$–reactive clones that used TRBJ2-2 or TRBJ2-4 were significantly more likely to have a larger clone size than those that did not (Fig S4D and E). Conversely, a negative correlation approaching statistical significance was found between TRBJ2-7 gene usage and clone size (Fig S4F). Increased TCR signaling has also been shown to impact cell fate by promoting effector function in acute LCMV infection and a RIP-OVA model of T1D (King et al, 2012), so we examined the phenotypic distributions of islet IGRP$_{206-214}$–reactive CD8 T cells expressing each major TCR *β* J gene. In parallel with their increased TCR signaling module scores, islet IGRP$_{206-214}$–reactive CD8 T cells expressing TRBJ2-2 also had a significantly increased

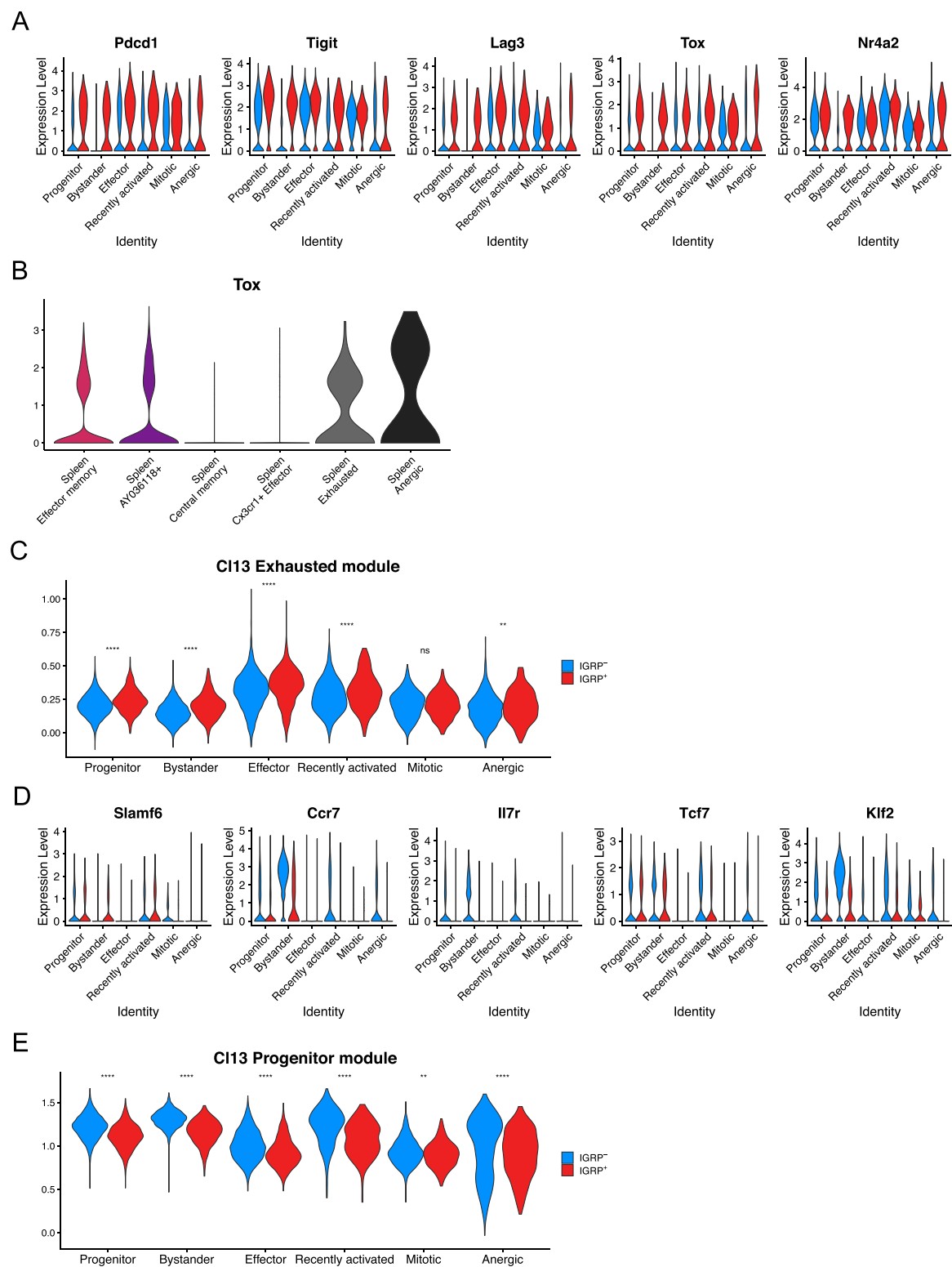

**Figure 3. IGRP$_{206-214}$–reactive CD8 T cells exhibit characteristics of increased antigen exposure compared with IGRP$_{206-214}$–nonreactive CD8 T cells.**
**(A)** Violin plots showing expression of key coinhibitory receptors, memory CD8 T cell-associated cell surface markers, and transcription factors regulating CD8 T-cell exhaustion. **(B)** Violin plot showing expression of *Tox* in spleen clusters. **(C)** Violin plot showing module scores of the top 100 differentially expressed genes from LCMV Clone 13 Exhausted CD8 T cells in islet CD8 T cells. Blue, IGRP$_{206-214}$–nonreactive CD8 T cells. Red, IGRP$_{206-214}$–reactive CD8 T cells. **(A, D)** As in (A) but showing markers of CD8 T-cell stemness. **(C, E)** As in (C), but showing module scores derived from the top 100 differentially expressed genes from LCMV Clone 13 Progenitor CD8 T cells. **P < 0.01, ****P < 0.0001 by the Wilcoxon test.

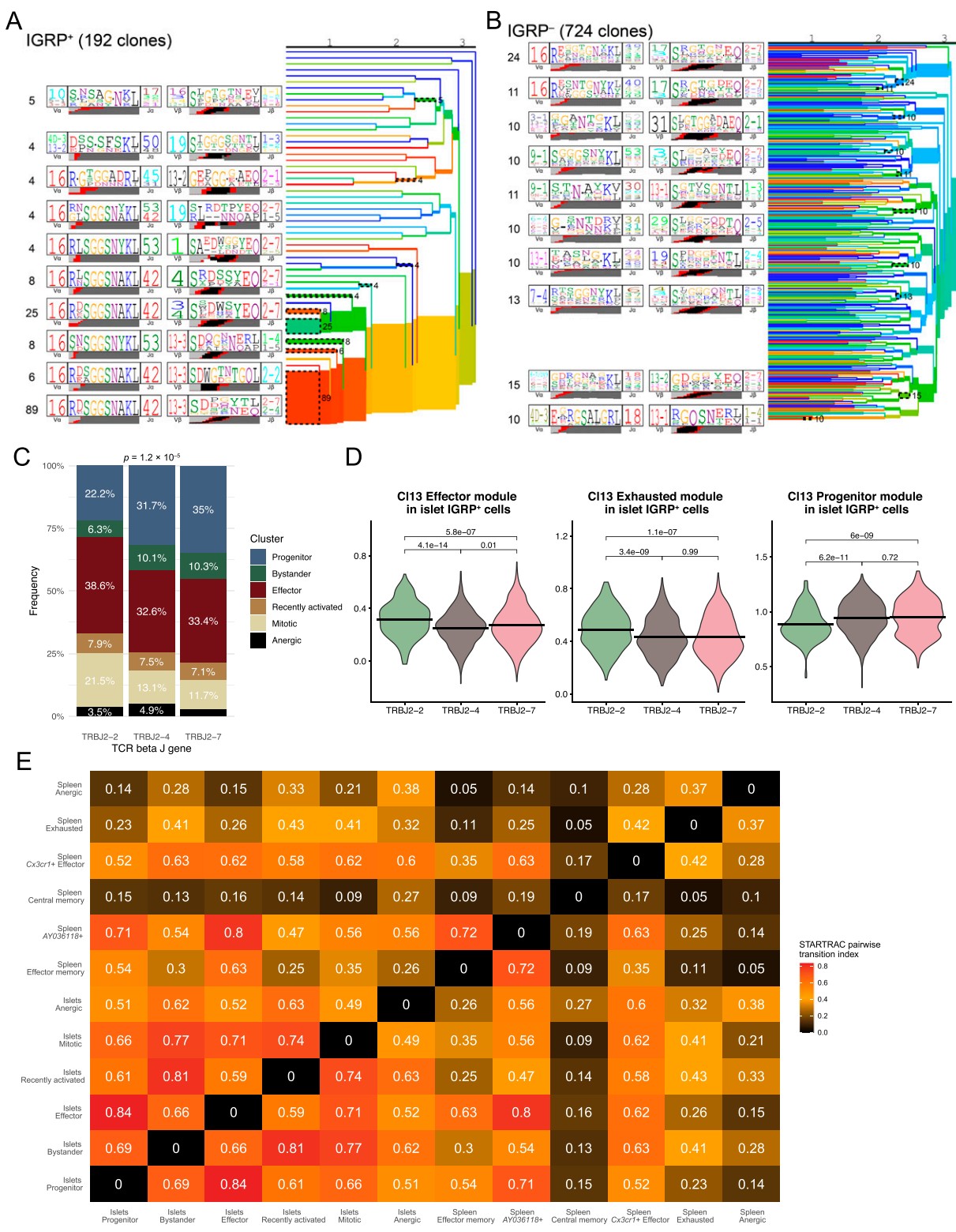

**Figure 4. Clonal analysis of autoreactive CD8 T cells in islets and spleen.**

**(A)** Clonal motif clustering generated by TCRdist of paired $\alpha$ and $\beta$ chain CDR3 sequences from IGRP$_{206-214}$–reactive CD8 T cells with at least two cells per clone. Colors underneath each motif denote origin of each amino acid: light gray, V gene; black, D gene; red, N nucleotide; dark gray, J gene. **(A, B)** As in (A), but from IGRP$_{206-214}$–nonreactive CD8 T cells. **(C)** Bar plot showing cluster distribution of islet IGRP$_{206-214}$–reactive CD8 T cells using the three dominant TCR $\beta$ J genes. **(D)** Violin plots showing LCMV Cl13 module scores among islet IGRP$_{206-214}$–reactive cells grouped by TCR $\beta$ J gene usage. **(E)** Heat map showing STARTRAC pairwise transition index of IGRP$_{206-214}$–reactive CD8 T cells from islets and spleens. Higher values indicate greater predicted differentiation potential between pairs of clusters.

frequency of effector cells compared with TRBJ2-4– and TRBJ2-7–expressing cells (Fig 4C). Accordingly, islet IGRP$_{206-214}$–reactive CD8 T cells expressing TRBJ2-2 had significantly higher LCMV Cl13 Effector and Exhausted module scores, and significantly lower Cl13 Progenitor Module scores, than those expressing either TRBJ2-4 or TRBJ2-7 (Fig 4D). Moreover, phenotypic differences among islet IGRP$_{206-214}$–reactive CD8 T cells were also mirrored in the spleen, with TRBJ2-2 usage correlating with an increased frequency of *Cx3cr1$^+$* effector cells compared with TRBJ2-4 or TRBJ2-7 gene usage (Fig S4G). These findings demonstrate that although IGRP$_{206-214}$–reactive CD8 T cell clones exhibit highly conserved CDR3 amino acid motifs, particularly in the TCR *α* chain, variable use of TCR *β* J genes may impact clonal expansion and effector potential of these cells.

Finally, we applied our scTCR-seq data to the clusters identified by scRNA-seq to predict phenotypic transitions among autoreactive CD8 T cells in the islets and spleen. We used the R package STARTRAC (Zhang et al, 2018) to identify clonal overlap among phenotypic clusters, quantifying overlap using the STARTRAC pairwise transition index (Fig 4C). Given that we did not sequence IGRP$_{206-214}$–negative cells from the spleen as these would not necessarily have diabetogenic potential, we performed this analysis only on IGRP$_{206-214}$–reactive cells from the islets and spleen. In general, we found that islet clusters exhibited high intercluster transition, likely the result of an immunostimulatory environment leading to active CD8 T-cell differentiation. The highest pairwise transition index was between the islet progenitor and islet effector clusters, in line with our previous work showing that progenitor cells give rise to effector cells under the influence of IL-27 signaling (Ciecko et al, 2021). Mitotic cells exhibited strong clonal overlap with all islet clusters except the anergic cluster, which in turn had the most clonal overlap with the bystander and recently activated clusters. This may imply that CD8 T-cell anergy in the islets tends to occur early, following priming and before clonal proliferation. Similar trends were found when we performed STARTRAC pairwise transition index analysis on IGRP$_{206-214}$–negative cells from the islets (Fig S4H): the progenitor and effector clusters exhibited strong pairwise transition scores, as did the anergic and recently activated clusters; on the other hand, bystander cells had low predicted transition with all clusters except the anergic cluster.

Effector memory, central memory, exhausted, and anergic IGRP$_{206-214}$–positive cells in the spleen had lower transition indices, indicative of more quiescent phenotypes with lower differentiation potential (Fig 4C). Central memory cells, in particular, had an extremely low transition index, in line with their predominantly non-expanded clonal composition (Fig S3C). Of note, although effector memory cells had relatively low transition indices on average, there were noticeable higher transition indexes between these cells and spleen *AY036118$^+$*, islet progenitor, and islet effector cells. In contrast, the *AY036118$^+$* and *Cx3cr1$^+$* effector clusters had high transition indices in general, suggesting that these cells are more phenotypically malleable. The spleen *AY036118$^+$* cluster and the islet effector cluster had a particularly high pairwise transition index, suggesting that the former may serve as an intermediate stage in memory formation from islet effector cells. Collectively, these data show that IGRP$_{206-214}$–reactive CD8 T cell clones have constituent cells in both the islets and spleen and raise the possibility that effector cells from the islets may recirculate and generate memory

cells in the spleen, as has been previously suggested (Chee et al, 2014).

# Discussion

In this study we used paired scRNA-seq and scTCR-seq to investigate the phenotypic and clonal heterogeneity of IGRP$_{206-214}$–reactive and –nonreactive diabetogenic CD8 T cells in the islets and spleens of the NOD murine model of T1DM. Our work demonstrates that IGRP$_{206-214}$–reactive CD8 T cells are highly restricted in terms of TCR gene usage, with the vast majority of these cells exhibiting TRAV16N–TRAJ42–TRBV13-3 TCR gene pairings at the single-cell level. These findings suggest that autoreactive T cells may inherently have restricted TCR repertoires as only specific TCR gene pairings are able to successfully slip through the net of thymic selection. This confirms and expands upon previous findings from CD8 T cell lines generated from islet-infiltrating CD8 T cells, which used anchor PCR to identify unpaired TCR *α* and *β* chain sequences (Santamaria et al, 1995). Of note, another study found that IGRP$_{206-214}$–reactive CD8 T cells from the islets of NOD mice exhibit avidity maturation at the population level, with V*α*17.6 (TRAV16*04), then V*α*17.4 (TRAV16*01), and finally V*α*17.5 (TRAV16*02) TCR *α* chain V genes being selected for in most of the IGRP$_{206-214}$–reactive CD8 T cells (Han et al, 2005b). Our data instead found a highly selective use of the TRAV16N gene, although some of this homogeneity may be due to the fact that the 10x Genomics Cell Ranger pipeline is limited in its ability to distinguish between all TRAV16 gene variants. It is therefore possible that the TCR *α* V gene variants previously reported by Han et al are present in our data. Alternatively, it may be possible that differences in the TCR repertoire of IGRP$_{206-214}$–reactive CD8 T cells may vary in NOD mice based upon their housing and diet; it has been shown that the gut microbiome impacts both the onset of T1DM in NOD mice (Tai et al, 2016) as well as T-cell maturation during thymic selection (Ennamorati et al, 2020; Cheng et al, 2021; Zegarra-Ruiz et al, 2021). It is plausible that some peptides from commensal microbiota may directly impact the development of IGRP$_{206-214}$–reactive CD8 T cells in the thymus by being presented on dendritic cells, as Tai et al demonstrated that the Mgt protein from Fusobacteria acts as an IGRP mimic and is capable of activating peripheral IGRP$_{206-214}$–reactive NY8.3 TCR transgenic CD8 T cells in vivo by oral gavage.

Our data expand upon previous evidence by Han et al due to technological advancements in sequencing technology. Using scTCR-seq, our study was able to identify TCR gene restriction not only in the TCR *α* V gene but also in the *α* J, *β* V, and *β* J genes at the single-cell level; moreover, we demonstrate that this holds true across several thousand diabetogenic CD8 T cells, all of which come from endogenous T cell repertoires rather than transgenic T cells or cultured cell lines. These novel findings therefore demonstrate that TCR gene usage in IGRP$_{206-214}$–reactive CD8 T cells is even more restricted than previously thought. Evidence of restricted TCR gene usage in IGRP$_{206-214}$–reactive CD8 T cells may also warrant the investigation of TCR gene usage in T cells reactive to other commonly targeted *β* cell epitopes, such as insulin (Amdare et al, 2021), at the single-cell level; these data parallel findings of restricted TCR

repertoires among autoreactive T cells found in other autoimmune disorders such as experimental autoimmune encephalomyelitis (Acha-Orbea et al, 1988; Urban et al, 1988; Sakai et al, 1989), polymyositis (Mantegazza et al, 1993), rheumatoid arthritis (Williams et al, 1992; Sharma et al, 2021), thyroiditis (Martin et al, 1999), and primary biliary cirrhosis (Moebius et al, 1990). It would also be valuable to apply scTCR-seq to CD8 T cells targeting other auto-antigens in the islets of NOD mice to see if this trend holds for CD8 T cells targeting other antigens; although we studied IGRP$_{206-214}$–nonreactive CD8 T cells in bulk in this study, we did not separately sort and sequence CD8 T cells targeting specific autoantigens other than IGRP$_{206-214}$ because of technical limitations of cell recovery. Restricted TCR gene usage may therefore be a fundamental property conserved among autoimmune disorders, as only certain genes may encode TCR structures that bind a given self-antigen strongly enough to overcome positive selection but weakly enough to evade negative selection in the thymus. These findings provide a window of opportunity to therapeutically target autoreactive T cells via selective depletion, although previous work performed using epitope-specific TCR altered peptide ligands has shown there are critical nuances to these approaches (Han et al, 2005a).

In addition to restricted TCR gene usage, IGRP$_{206-214}$–reactive CD8 T clones showed strong TCR motif overlap among many clones. The majority (70.3%) of IGRP$_{206-214}$–reactive clones exhibited a highly conserved "RDSG" TCR $\alpha$ chain CDR3 motif, a result of both restricted TRAV16-N/TRAJ42 gene usage as well as N nucleotide additions within the CDR3. We also identified three consensus $\beta$ chain CDR3 motifs arising from the three most dominant TCR $\beta$ J genes: TRBJ2-2, TRBJ2-4, and TRBJ2-7. Expression of TRBJ2-2 significantly correlated with increased clone size, increased TCR signaling module scores, increased LCMV Cl13 Effector and Exhausted module scores among islet CD8 T cells, and increased frequencies of effector cells in the islets and spleen. Collectively, these analyses reveal striking TCR $\alpha$ chain CDR3 motif similarity among IGRP$_{206-214}$–reactive CD8 T cells clones in the islets and spleen while also suggesting that the TCR $\beta$ chain CDR3 may be slightly less restricted because of TCR $\beta$ J gene heterogeneity; expression of TCR $\beta$ J genes may also impact clonal expansion and effector function of IGRP$_{206-214}$–reactive clones during the autoimmune response in NOD mice. Of note, our observations mirror TCR-seq data from human T1D patients, whose IGRP$_{265-273}$–specific CD8 T cells also exhibited highly restricted TCR $\alpha$ gene usage but less restricted TCR $\beta$ gene usage (Fuchs et al, 2017).

Despite the remarkable TCR similarly present among IGRP$_{206-214}$–reactive clones, we found that IGRP$_{206-214}$–reactive CD8 T-cell clones are extremely heterogeneous and rarely overlap (~5% overlap for two groups of 10 mice) between biological organisms. This is similar to our previous findings that there is minimal clonal overlap in the virus-specific CD4 T-cell pool among several genetically identical organisms (Khatun et al, 2021). As these clones are highly restricted in terms of germline TCR gene usage, the clonal diversity in the IGRP-specific CD8 T-cell repertoire arises both from combinatorial diversity of V and J gene pairings, junctional diversity of N nucleotides added between TCR genes during V(D)J recombination, and paired diversity of $\alpha$ chain and $\beta$ chain sequences for each cell. These details may have hitherto been obscured but are now coming to light given the resolution afforded by scTCR-seq. Application of this technique to other autoimmune

disorders may unveil similar trends in both clonal restriction and heterogeneity to those we have identified among diabetogenic T cells.

Using scRNA-seq, we found several distinct phenotypes of diabetogenic CD8 T cells in the islets. Four of these clusters (bystander, progenitor, mitotic, and effector) were recently identified by our laboratory (Ciecko et al, 2021), whereas the recently activated cluster and anergic cluster were identified by our current work. We also identified six clusters of IGRP$_{206-214}$–reactive CD8 T cells in the spleen. Effector memory and *AY036118*$^+$ cells had a similar memory-like phenotype, suggesting that autoreactive memory CD8 T cells can be found in lymphoid tissues. Regulon analysis suggested that these memory T cell subsets may depend on the TF Zeb2. Generation of diabetogenic memory T cells is a critical barrier to the use of islet transplants (Ehlers & Rigby, 2015), so identification and inhibition of pathways that promote memory formation may ultimately be of clinical benefit. Splenic *Cx3cr1*$^+$ effector cells were similar to islet effector CD8 T cells as both populations expressed the cytotoxic molecule *Gzmb*; however, the former expressed lower levels of coinhibitory receptors such as *Pdcd1*, *Lag3*, and *Tigit* and higher levels of the effector marker *Klrg1* (Joshi et al, 2007; Zander et al, 2019). This gene expression profile suggests that *Cx3cr1*$^+$ effector cells in the spleen have a more cytotoxic phenotype whereas effector cells in the islets exhibit characteristics of exhausted T cells. Although we did not find a discrete exhausted subset of CD8 T cells in the islets, we did identify a small number of exhausted cells in the spleen. These cells were characterized by expression of *Cxcr6*, *Pdcd1*, *Tigit*, and *Eomes* and were reminiscent of both CXCR6$^+$ PD-1$^{hi}$ exhausted CD8 T cells from LCMV Cl13 infection (Zander et al, 2019) as well as TIGIT$^+$ Eomes$^{hi}$ CD8 T cells found in the peripheral blood of T1D patients (Wiedeman et al, 2020). Finally, we identified central memory and anergic cells in the spleen, both of which had limited clonal expansion and predicted differentiation potential. Although the splenocytes we sequenced were sorted on CD44$^+$ cells, it may be possible that the splenic central memory cluster may also contain some naïve CD8 T cells given that naïve and memory CD8 T cells share similar phenotypic and epigenetic profiles (Schauder et al, 2021). Collectively, our data therefore identify previously under-appreciated phenotypic heterogeneity among diabetogenic CD8 T cells and show that several different CD8 T-cell populations exist in the islets and spleens of NOD mice during the onset of T1D.

Although our analyses identified variable clonal overlap between islet effector clusters and splenic effector and memory-like populations, there are still unresolved questions regarding the differentiation process of these cells. The relationship between islet effector cells and splenic *Cx3cr1*$^+$ effector cells is of particular interest. Although it is possible that the former population could up-regulate the chemokine receptor CX$_3$CR1 to exit the islets and enter the circulation to eventually travel to the spleen and become *Cx3cr1*$^+$ effector cells, islet effector cells express the inhibitory coreceptors *Pdcd1* and *Tigit*, typically expressed by exhausted cells, whereas these genes are not highly expressed in *Cx3cr1*$^+$ effector cells. Moreover, islet effector cells express *Tox*, a master regulator of T-cell exhaustion (Alfei et al, 2019; Khan et al, 2019; Scott et al, 2019; Seo et al, 2019; Yao et al, 2019), whereas this key marker is not expressed by splenic *Cx3cr1*$^+$ effector cells. Alternatively, splenic *Cx3cr1*$^+$ effector cells may travel from pancreatic lymph nodes,

where they are activated but have not yet migrated to pancreatic islets. Single-cell assay for transposase-accessible chromatin (scATAC-seq) may therefore be useful to map the epigenetic landscape of these two populations and splenic exhausted cells to determine whether transdifferentiation is possible. scATAC-seq was recently used to study autoreactive CD8 T cells found in the peripheral blood of T1D patients (Abdelsamed et al, 2020) and would provide a valuable comparison with future data from murine diabetogenic CD8 T cells. High resolution sequencing techniques such as combined scRNA-seq and scTCR-seq, as used in this work, may therefore shed new light on the phenotypic and clonotypic nuances of T cells in T1D and other autoimmune disorders.

# Materials and Methods

### Mice

NOD/ShiLtJ (NOD) mice were procured from The Jackson Laboratory and maintained at the Medical College of Wisconsin (MCW). All mice were bred and maintained under the guidelines of the MCW Institutional Animal Care and Use Committee (IACUC).

### Isolation of cells from pancreatic islets and spleen

Pancreatic islets were isolated as previously described (Foda et al, 2020). Briefly, pancreatic islets were isolated by perfusing the pancreas via the common bile duct with collagenase P solution (0.5 units/ml). The inflated pancreata were incubated for 16 min at 37°C, agitated, and washed with Hank's balanced salt solution (Sigma-Aldrich) plus 2% fetal bovine serum. Islets were hand-picked from the pancreas digestion slurry and dissociated in non-enzymatic cell dissociation buffer. Islet cell suspensions were resuspended in RPMI and incubated at 37°C for 1 h. Final islet single cell suspensions were washed and resuspended in RPMI. Spleens were harvested and mechanically processed with a 70 $\mu$m filter to obtain a single-cell suspension. Red blood cell lysis was performed using ACK lysing buffer (Lonza). Splenocytes were enriched for CD8 T cells using magnetic bead-based negative selection (CD8 mouse T-cell isolation kit; Miltenyi Biotec). Final spleen single-cell suspensions were washed and resuspended in RPMI. Islets and spleens were each pooled from 10 mice for both scRNA-seq experiments.

### Cell sorting

Spleen and islet single cell suspensions were incubated with 5 $\mu$g/ml of FC block (anti-mouse CD16/CD32 clone 2.4G2; Bio X Cell). Cells were simultaneously incubated for 15 min at room temperature with MHC class I ($K^d$) tetramers loaded with mimotope peptide NRP-V7 (Trudeau et al, 2003). Cells were then stained with antibodies against CD45.1 (A20), CD8 (53-6.7), CD44 (IM7), and lineage markers (Lin) CD4 (RM4-5), B220 (RA3-6B2), and CD11b (M1/70) for 30 min at 4°C and were then washed and passed through a 30 $\mu$m filter. Single CD45.1$^+$ Lin$^-$CD8$^+$ NRP-V7 Tet$^+$ cells were sorted from spleen and islet samples and single CD45.1$^+$ Lin$^-$CD8$^+$ NRP-V7 Tet$^-$ cells were sorted from islet samples using a FACSAria II cell sorter (BD Biosciences).

### Single-cell RNA sequencing

Cells were loaded into the 10x Chromium Controller (10x Genomics) for barcoding. scRNA-seq and scTCR-seq libraries were prepared using the Chromium Single Cell 5′ v3 Reagent Kit according to the manufacturer's protocol. Libraries were quantified using a KAPA library quantification kit (Roche Sequencing) and loaded onto an Illumina NextSeq 500 sequencer. A NextSeq 500/550 High Output Kit v2.5 (150 cycles) (20024907; Illumina) with 26 cycles for read 1, 91 cycles for read 2, and 8 cycles for the index read was used for scRNA-seq libraries from the first experiment (Round 1). A NextSeq 500/550 High Output Kit v2.5 (300 cycles) (20024908; Illumina) with 151 cycles for read 1, 151 cycles for read 2, and 8 cycles for the index read was used for scTCR-seq libraries from Round 1. A NextSeq 500/550 High Output Kit v2 (300 cycles) (20024908; Illumina) with 149 cycles for read 1, 149 cycles for read 2, and 10 cycles for index reads was used for pooled scRNA-seq and scTCR-seq libraries from the second experiment (Round 2). Raw sequencing data were downloaded from Illumina BaseSpace and demultiplexed using the "mkfastq" and "count" functions in Cell Ranger (v 6.0) (10x Genomics). scRNA-seq FASTQ files were aligned to the mm10 reference transcriptome (10x Genomics) using the "count" function, whereas scTCR-seq FASTQ files were aligned to the GRCm38 V(D)J reference transcriptome (10x Genomics). 7,302, 1,373, and 3,708 cells were recovered from islet IGRP$^-$, islet IGRP$^+$, and spleen IGRP$^+$ Round 1 scRNA-seq samples, respectively. 5,550, 1,011, and 2,959 cells were recovered from islet IGRP$^-$, islet IGRP$^+$, and spleen IGRP$^+$ Round 1 scTCR-seq samples, respectively. 9,877, 1,889, and 2,669 cells were recovered from islet IGRP$^-$, islet IGRP$^+$, and spleen IGRP$^+$ Round 2 scRNA-seq samples, respectively. 8,495, 1,585, and 2,361 cells were recovered from islet IGRP$^-$, islet IGRP$^+$, and spleen IGRP$^+$ Round 2 scTCR-seq samples, respectively.

### Single-cell RNA sequencing analysis

Data analysis was performed in R (v 4.0.2) using the package Seurat (v 4.0.3) (Stuart et al, 2019), with the package tidyverse (v 1.2.1) (Wickham et al, 2019) used to organize data and the package ggplot2 (v 3.2.1) used to generate figures. scRNA-seq data were filtered to keep cells with a low percentage of mitochondrial genes in the transcriptome (<5%) and between 200 and 2000 (spleen) or 200 and 3,500 (islets) unique genes to exclude apoptotic cells, low quality cells, and doublets. Cell cycle scores were regressed when scaling gene expression values and T cell receptor genes were regressed during scTransform integration (Hafemeister & Satija, 2019) and clustering, which was performed using 50 principal components with the Louvain algorithm and visualized with UMAP plots. Islet and spleen samples were integrated and clustered separately to avoid overfitting phenotypes. From the islet data, one cluster of 237 pancreatic exocrine cells was excluded before analysis. A clone was defined as a group of cells sharing the same TCR $\alpha$ and $\beta$ chain CDR3 nucleotide sequences. STARTRAC (v 0.1.0) (Zhang et al, 2018) was used to calculate clonal metrics of differentiation potential. TCRdist (Dash et al, 2017) was used to perform CDR3 motif analysis. Chord diagrams were generated with circlize (v 0.4.8) (Gu et al, 2014), Venn diagrams with VennDiagram (v 1.6.20) (Chen & Boutros, 2011), and UpSet diagrams with ggupset (v 0.3.0). Module scores were

calculated with stated gene lists using Seurat's AddModuleScore function with default parameters. Regulon analysis was performed using DoRothEA (v 1.0.1) (Garcia-Alonso et al, 2019; Holland et al, 2020a; 2020b) with default settings; only TFs with confidence ratings of A, B, or C (validated ChIP-seq binding and present in DoRothEA curated resources) were used for analysis.

### Statistical analysis

Statistical tests were performed in R (v 4.0.2). Violin plots were compared pairwise using the Wilcoxon test with Holm–Sidak correction using the ggpubr package (v 0.4.0). Bar graphs and logistic regression plots were analyzed using the chi-squared test.

### Data and code availability

Single-cell RNA sequencing and single-cell TCR sequencing data from this article are accessible in the GEO database (accession number GSE200608). All other raw data and scripts are available from the corresponding author upon request.

## Supplementary Information

## Acknowledgements

This work is supported by National Institutes of Health (NIH) grants DK107541 (Yi-Guang Chen), DK121747 (Yi-Guang Chen), AI125741 (W Cui), AI148403 (W Cui), DK127526 (MY Kasmani), and DK118786 (AE Ciecko); by an American Cancer Society Research Scholar Grant (W Cui); and by an Advancing a Healthier Wisconsin Endowment Grant (W Cui). MY Kasmani and AK Brown are members of the Medical Scientist Training Program at the Medical College of Wisconsin (MCW), which is partially supported by a training grant from the National Institute of General Medical Sciences (NIGMS) (T32-GM080202). This research was completed in part with computational resources and technical support provided by the Research Computing Center at MCW. NRP-V7 tetramer was obtained through the NIH Tetramer Core Facility.

### Author Contributions

MY Kasmani: conceptualization, data curation, software, formal analysis, validation, investigation, visualization, and writing—original draft, review, and editing.
AE Ciecko: conceptualization, data curation, software, formal analysis, validation, investigation, visualization, and writing—original draft, review, and editing.
AK Brown: software, formal analysis, visualization, and writing—review and editing.
G Petrova: investigation.
J Gorski: formal analysis and writing—review and editing.
Y-G Chen: conceptualization, resources, supervision, funding acquisition, project administration, and writing—review and editing.
W Cui: conceptualization, resources, supervision, funding acquisition, project administration, and writing—review and editing.

### Conflict of Interest Statement

The authors declare that they have no conflict of interest.

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
