## [Reviewer comments · Life Science Alliance]

Life Science Alliance

Autoreactive CD8 T cells in NOD mice exhibit phenotypic heterogeneity but restricted TCR gene usage

Moujtaba Kasmani, Ashley Ciecko, Ashley Brown, Galina Petrova, Jack Gorski, Yi-Guang Chen, and Weiguo Cui
DOI: <https://doi.org/10.26508/lsa.202201503>

Corresponding author(s): Weiguo Cui, Versiti blood research Institute and Yi-Guang Chen, Medical College of Wisconsin

Review Timeline:

Submission Date:	2022-04-25
Editorial Decision:	2022-05-12
Revision Received:	2022-05-19
Editorial Decision:	2022-05-23
Revision Received:	2022-05-24
Accepted:	2022-05-25

Transaction Report:

May 12, 2022

Re: Life Science Alliance manuscript #LSA-2022-01503-T

Dr. Weiguo Cui
Blood Research Institute, Versiti Wisconsin
Milwaukee, WI

Dear Dr. Cui,

Thank you for submitting your manuscript entitled "Autoreactive CD8 T cells in NOD mice exhibit phenotypic and clonal heterogeneity despite restricted TCR gene usage" to Life Science Alliance. The manuscript was assessed by expert reviewers, whose comments are appended to this letter. We invite you to submit a revised manuscript addressing the Reviewer comments.

Thank you for this interesting contribution to Life Science Alliance. We are looking forward to receiving your revised manuscript.

Sincerely,

B. MANUSCRIPT ORGANIZATION AND FORMATTING:

Reviewer #1 (Comments to the Authors (Required)):

In this manuscript, Cui and colleagues make single-cell RNAseq and TCR analysis of CD8 T cell populations existing in islets and spleen of pre-diabetic NOD mice. To this end, they isolate the T cell populations according to their binding or not to tetramers of the peptide IGRP(206-214), as one of the most important epitopes mediating autoimmunity. The authors find strong conservation of TCRalpha-TCRbeta pairings and of V and J regions in both TCR subunits but quite a big diversity in CDR3 hypervariable regions. Out of the scRNAseq data, the authors extract information about the phenotype of the cells infiltrating the islets and the spleen in terms of being stimulated, differentiated to effector cells, memory, etc.

I find the work relatively limited in terms of the scope of the conclusions reached.

First, the selection of CD8 T cells by the IGRP(206-214) tetramer limits the analysis of potentially pathogenic T cells to that epitope and there is no information about the existence of other populations with other autoantigen specificities.

Second, the possible functional analysis is limited to the expression of a couple of genes that the authors classify as effector, memory, etc but without a functional validation of such behaviors. Indeed, there is no single experiment of stimulation *ex vivo* or *in vivo* to demonstrate that the classification of T cells according to the expression of those few markers is indeed functionally relevant.

Third, the analysis is carried out on pre-diabetic NOD mice but there is no idea on how long will it take for those mice to develop overt disease and how the detected populations change during early onset of disease. So, the future behavior of the T cell populations characterized by usage of V, J regions and CDR3 sequences is purely speculative.

Fourth, the restriction of V J usage by the single-used MHC tetramer could be expected due to the constrictions for binding of the TCR chains to the MHC tetramer. Therefore, it could be expected. A more interesting result would have been the analysis of common usage of V, J regions and the existence of conserved sequences in a global response, caused not by one but by different autoantigens.

In summary, the limitations posed by the single use of sequence analysis technologies without functional validation make the work quite inconclusive.

Reviewer #2 (Comments to the Authors (Required)):

In this manuscript, Kasmani and colleagues describe paired scRNAseq and scTCRseq of CD8 T cells isolated from the spleen and pancreatic islets of pre-diabetic NOD mice, a model for type 1 diabetes. The authors follow-up on their previous studies that had already described several key phenotypes within the CD8 T cell population at play in this model for autoimmune diabetes. Here, they added information about TCR specificity and clonality. In particular, they differentiated between IGRP206-214 reactive T cells and non-IGRP reactive ones. This particular epitope is a dominant antigen in the CD8 compartment of NOD mice, and allowed the authors to study a disease-relevant specificity in contrast to a mixture of CD8 cells, some of which may be bystanders, others reacting to less dominant islet antigens.

The authors describe differences in clonality and expansion of IGRP- vs. non-IGRP-reactive cells, and further describe the different subpopulations of cells they classified based on scRNAseq profiles. These include activated, mitotic, anergic, exhausted, 'progenitor' (memory) and bystander populations. The authors conclude from their analyses that a very limited number of TCR chain combinations account for most IGRP-reactive clones. Yet, exact clones (based on CDR3 sequences) are very rarely shared across groups of mice. The authors identify six clusters of islet CD8 phenotypes in this study, four of which they have described in earlier work, the other two being new (recently activated and anergic). The authors go on to discuss possible relationships between these phenotypes, and contrast their results from spleen cells with those found in islets.

Overall, this is a well conducted study, with objective description and interpretation of the data, and an extensive discussion. The body of work adds important and interesting insight to our understanding of T cell phenotypes in this model for type 1 diabetes. While some of the phenotypes and their progression from one subpopulation to another remain somewhat speculative in the absence of functional studies, the data is of value to the field.

I found nothing of concern with the data presentation or interpretation that would warrant revision, and I would deem the manuscript acceptable as is.

Reviewer #1

First, the selection of CD8 T cells by the IGRP(206-214) tetramer limits the analysis of potentially pathogenic T cells to that epitope and there is no information about the existence of other populations the other autoantigen specificities.

The Reviewer makes an important point about antigen specificity playing a major role in autoreactivity. We sequenced both IGRP₂₀₆₋₂₁₄-reactive and -nonreactive CD8 T cells from the islets of NOD mice for this very reason, especially since IGRP₂₀₆₋₂₁₄ has been a major epitope of interest for study. However, we did not sequence separate populations of CD8 T cells specific for other autoepitopes such as INS-B₁₅₋₂₃ as these populations are individually present at much lower frequencies than IGRP₂₀₆₋₂₁₄-reactive CD8 T cells. Therefore, this lack of data is due to technical limitations of cell recovery. Despite this, we were still able to identify altered phenotypic ratios between IGRP₂₀₆₋₂₁₄-reactive and polyclonal IGRP₂₀₆₋₂₁₄-nonreactive CD8 T cells. For example, we found that the IGRP₂₀₆₋₂₁₄-reactive CD8 T cells had significantly lower frequencies of bystander cells and significantly higher frequencies of effector cells compared to IGRP₂₀₆₋₂₁₄-nonreactive CD8 T cells (**Figure 2C**), likely as a result of increased antigen stimulation (**Figure 3A, C**). However, we acknowledge that this is a point that should be addressed in the main manuscript. We have therefore added points to the introduction (lines 89-90) and discussion (lines 479-484) emphasizing that this manuscript is primarily focused on IGRP₂₀₆₋₂₁₄-reactive CD8 T cells but that this leads to inherent limitations in the information gained from our studies.

Second, the possible functional analysis is limited to the expression of a couple of genes that the authors classify as effector, memory, etc but without a functional validation of such behaviors. Indeed, there is no a single experiment of stimulation ex vivo or in vivo to demonstrate that the classification of T cells according to the expression of those few markers is indeed functionally relevant

We appreciate the Reviewer's emphasis on biological validation. We and other groups have previously functionally characterized several of the subsets in this manuscript, specifically the bystander, progenitor, mitotic, and effector populations (Ciecko et al., 2021; Gearty et al., 2022; Hu et al., 2020); we have now specified this in the revised manuscript (lines 208-212). The two new populations we identified, recently activated and anergic, are both relatively small (approximately 5% of islet CD8 T cells by scRNA-seq) and may be difficult to distinguish by flow cytometry surface markers; running functional assays on these populations would therefore be

quite challenging technically. In addition, we would like to emphasize that the focus of our manuscript is not the functionality of these two new subsets, but rather the interplay between phenotypes and clonotypes of diabetogenic CD8 T cells. We hope that our work provides insight into these relationships and serves as a useful resource for the broad community of T1D researchers.

Third, the analysis is carried out on pre-diabetic NOD mice but there is no idea on how long will it take for those mice to develop overt disease and how the detected populations change during early onset of disease. So, the future behavior of the T cell populations characterized by usage of V, J regions and CDR3 sequences is purely speculative.

The Reviewer's point about kinetics is important. It is true that our samples are pooled from mice between 10 and 15 weeks of age. However, we do know that female NOD mice in the Medical College of Wisconsin colony typically start to develop T1D around 12 weeks of age, and the incidence reaches ~70% at 20 weeks of age. The mice used for this study were therefore on the threshold of developing T1D. Obtaining enough NOD mice that are exactly age-matched to within one week of age and performing these experiments multiple times would require significant further work. Although we believe this would be beyond the scope of our study, it would provide valuable insight into the kinetics of disease. Finally, we are unsure of what the Reviewer is referring to concerning "future behavior" of T cells based on TCR structure. Our analyses of TCR structure strictly draw correlations between gene usage and phenotypes of CD8 T cell clones in our data at the timepoint the experiments were conducted at. We do not attempt to predict future differentiation based on this information. On a more pragmatic note, it would be difficult to obtain data for CD8 T cells past the 15 week timepoint we used, as there would not be enough CD8 T cells that can be recovered from the remaining in the islets this late in the disease course. In short, we apologize if our writing appeared to state that our data was predictive rather than correlative, as we did not intend to suggest our data can predict future differentiation states of CD8 T cells; we would be happy to rephrase such sections of the manuscript if the Reviewer specifies any sentences that could be misinterpreted.

Fourth, the restriction of V J usage by the single-used MHC tetramer could be expected due to the constrictions for binding of the TCR chains to the MHC tetramer. Therefore, it could be expected. A more interesting result would have been the analysis of common usage of V, J regions and the existence of conserved sequences in a global response, caused not by one but

by different autoantigens.

The Reviewer raises an important point about establishing a baseline of what to expect from our data. Our scTCR-seq analyses of IGRP₂₀₆₋₂₁₄-reactive CD8 T cells show that these cells have very restricted TCR gene usage. It may be possible that such gene restriction could be normal for T cells regardless of what epitope they target. However, our scTCR-seq analysis of CD4 T cells targeting the GP66 epitope of lymphocytic choriomeningitis virus (LCMV) has shown that antiviral CD4 T cells targeting a single epitope are not restricted in terms of TCR gene usage, with GP66-specific CD4 T cells from each of 5 separate mice expressing at least 40 different TCR alpha V genes (Khatun et al., 2021). We have performed a similar analysis on antiviral CD8 T cells targeting the GP33 epitope of LCMV (unpublished data) and have also found extensive heterogeneity in TCR gene usage among GP33-specific antiviral CD8 T cells (see Figure 1 below). These data suggest that the restricted TCR gene usage we observed in IGRP₂₀₆₋₂₁₄-reactive CD8 T cells is not necessarily seen among all epitope-specific CD8 T cell repertoires. Given that TCR gene restriction has previously been observed in T cells in type 1 diabetes (Santamaria et al., 1995) and other autoimmune disorders such as experimental autoimmune encephalomyelitis (Sakai et al., 1989; Urban et al., 1988; Acha-Orbea et al., 1988), polymyositis (Mantegazza et al., 1993), rheumatoid arthritis (Williams et al., 1992; Sharma et al., 2021), thyroiditis (Martin et al., 1999), and primary biliary cirrhosis (Moebius et al., 1990), it may be possible that autoreactive T cells are inherently restricted in their TCR gene usage via thymic selection. However, our scTCR-seq analyses have allowed us to demonstrate this restriction in gene usage at the single-cell level using several thousand endogenous primary CD8 T cells rather than cell lines or transgenic T cells, a level of resolution not previously possible.

Regarding the Reviewer's comment about examining autoreactive CD8 T cells globally, we believe it is unlikely that we would find many conserved TCR genes given the extensive heterogeneity in TCR gene usage we see just among pooled diabetogenic IGRP₂₀₆₋₂₁₄-nonreactive CD8 T cells from the pancreatic islets. However, it may be possible that there may be some key signatures of either TCR gene usage or transcriptional profiles shared among various autoreactive CD8 T cells in models of T1D or even other autoimmune disorders. Although such findings would be intriguing, these would require extensive experiments and analyses that we feel are beyond the scope of this study. Nonetheless, we appreciate the Reviewer's feedback and curiosity about potential clonal overlap among broader groups of autoreactive T cells.

Figure 1. Chord diagrams of LCMV GP33-specific CD8 T cells isolated from two separate mice.

Reviewer #2

I found nothing of concern with the data presentation or interpretation that would warrant revision, and I would deem the manuscript acceptable as is.

We would like to thank the Reviewer for taking the time to review the manuscript thoroughly, and we appreciate their positive feedback.

References

- Acha-Orbea, H., D.J. Mitchell, L. Timmermann, D.C. Wraith, G.S. Tausch, M.K. Waldor, S.S. Zamvil, H.O. McDevitt, and L. Steinman. 1988. Limited heterogeneity of T cell receptors from lymphocytes mediating autoimmune encephalomyelitis allows specific immune intervention. *Cell*. 54:263–273. doi:10.1016/0092-8674(88)90558-2.
- Ciecko, A.E., D.M. Schauder, B. Foda, G. Petrova, M.Y. Kasmani, R. Burns, C. Lin, W.R. Drobyski, W. Cui, and Y. Chen. 2021. Self-Renewing Islet TCF1+ CD8 T Cells Undergo IL-27–Controlled Differentiation to Become TCF1– Terminal Effectors during the Progression of Type 1 Diabetes. *J. Immunol.* 121747:ji2100362. doi:10.4049/jimmunol.2100362.
- Gearty, S. V., F. Dündar, P. Zumbo, G. Espinosa-Carrasco, M. Shakiba, F.J. Sanchez-Rivera, N.D. Soggi, P. Trivedi, S.W. Lowe, P. Lauer, N. Mohibullah, A. Viale, T.P. DiLorenzo, D. Betel, and A. Schietinger. 2022. An autoimmune stem-like CD8 T cell population drives type 1 diabetes. *Nature*. 602:156–161. doi:10.1038/s41586-021-04248-x.
- Hu, H., P.N. Zakharov, O.J. Peterson, and E.R. Unanue. 2020. Cytocidal macrophages in symbiosis with CD4 and CD8 T cells cause acute diabetes following checkpoint blockade of PD-1 in NOD mice. *Proc. Natl. Acad. Sci.* 117:31319–31330. doi:10.1073/pnas.2019743117.
- Khatun, A., M.Y. Kasmani, R. Zander, D.M. Schauder, J.P. Snook, J. Shen, X. Wu, R. Burns, Y.-G. Chen, C.-W. Lin, M.A. Williams, and W. Cui. 2021. Single-cell lineage mapping of a diverse virus-specific naive CD4 T cell repertoire. *J. Exp. Med.* 218. doi:10.1084/jem.20200650.
- Mantegazza, R., F. Andreetta, P. Bernasconi, F. Baggi, J.R. Oksenberg, O. Simoncini, M. Mora, F. Cornelio, and L. Steinmant. 1993. Analysis of T Cell Receptor Repertoire of Muscle-infiltrating T Lymphocytes in Polymyositis. *J. Clin. Invest.* 91:2880–2886.
- Martin, A., G. Barbesino, and T.F. Davies. 1999. T-Cell Receptors and Autoimmune Thyroid Disease – Signposts for T-Cell-Antigen Driven Diseases. *Int. Rev. Immunol.* 18:111–140. doi:10.3109/08830189909043021.
- Moebius, U., M. Manns, G. Hess, G. Kober, K.M. zum Büschenfelde, and S.C. Meuer. 1990. T cell receptor gene rearrangements of T lymphocytes infiltrating the liver in chronic active hepatitis B and primary biliary cirrhosis (PBC): Oligoclonality of PBC-derived T cell clones. *Eur. J. Immunol.* 20:889–896. doi:10.1002/eji.1830200426.
- Sakai, K., S.S. Zamvil, D.J. Mitchell, S. Hodgkinson, J.B. Rothbard, and L. Steinman. 1989. Prevention of experimental encephalomyelitis with peptides that block interaction of T cells with major histocompatibility complex proteins. *Proc. Natl. Acad. Sci. U. S. A.* 86:9470–9474. doi:10.1073/pnas.86.23.9470.

- Santamaria, P., T. Utsugi, B.J. Park, N. Averill, S. Kawazu, and J.W. Yoon. 1995. Beta-cell-cytotoxic CD8+ T cells from nonobese diabetic mice use highly homologous T cell receptor alpha-chain CDR3 sequences. *J. Immunol.* 154:2494–503.
- Sharma, R.K., S. V. Boddul, N. Yoosuf, S. Turcinov, A. Dubnovitsky, G. Kozhukh, F. Wermeling, W.W. Kwok, L. Klareskog, and V. Malmström. 2021. Biased TCR gene usage in citrullinated Tenascin C specific T-cells in rheumatoid arthritis. *Sci. Rep.* 11:1–8. doi:10.1038/s41598-021-04291-8.
- Urban, J.L., V. Kumar, D.H. Kono, C. Gomez, S.J. Horvath, J. Clayton, D.G. Ando, E.E. Sercarz, and L. Hood. 1988. Restricted use of T cell receptor V genes in murine autoimmune encephalomyelitis raises possibilities for antibody therapy. *Cell.* 54:577–592. doi:10.1016/0092-8674(88)90079-7.
- Williams, W. V., Q. Fang, D. Demarco, J. VonFeldt, R.B. Zurier, and D.B. Weiner. 1992. Restricted heterogeneity of T cell receptor transcripts in rheumatoid synovium. *J. Clin. Invest.* 90:326–333. doi:10.1172/JCI115866.

May 23, 2022

RE: Life Science Alliance Manuscript #LSA-2022-01503-TR

Dr. Weiguo Cui
Blood Research Institute, Versiti Wisconsin
Milwaukee, WI

Dear Dr. Cui,

Thank you for submitting your revised manuscript entitled "Autoreactive CD8 T cells in NOD mice exhibit phenotypic heterogeneity but restricted TCR gene usage". We would be happy to publish your paper in Life Science Alliance pending final revisions necessary to meet our formatting guidelines.

- please add ORCID ID for both corresponding authors-you should have received instructions on how to do so
- please use the [10 author names, et al.] format in your references (i.e. limit the author names to the first 10)

A. FINAL FILES:

B. MANUSCRIPT ORGANIZATION AND FORMATTING:

Sincerely,

May 25, 2022

RE: Life Science Alliance Manuscript #LSA-2022-01503-TRR

Weiguo Weiguo Cui
Versiti blood research Institute
Immunology
8727 Watertown Plank Road
Versiti Blood Research Institute
Milwaukee, WI 53226

Dear Dr. Cui,

Thank you for submitting your Research Article entitled "Autoreactive CD8 T cells in NOD mice exhibit phenotypic heterogeneity but restricted TCR gene usage". It is a pleasure to let you know that your manuscript is now accepted for publication in Life Science Alliance. Congratulations on this interesting work.

DISTRIBUTION OF MATERIALS:

Again, congratulations on a very nice paper. I hope you found the review process to be constructive and are pleased with how the manuscript was handled editorially. We look forward to future exciting submissions from your lab.

Sincerely,
